# Learning from uncertain concepts via test time interventions

**Ivaxi Sheth** [*]
Mila, ÉTS Montréal

**Aamer Abdul Rahman**
Mila, ÉTS Montréal

**Laya Rafiee**
Concordia University

**Mohammad Havaei**
ICH

**Samira Ebrahimi Kahou**
Mila, ÉTS Montréal, CIFAR AI Chair

## Abstract

With neural networks applied to safety-critical applications, it has become increasingly important to understand the defining features of decision-making. Therefore, the need to uncover the black boxes to rational representational space of these neural networks is apparent. Concept bottleneck model (CBM) encourages interpretability by predicting human-understandable concepts. They predict concepts from input images and then labels from concepts. Test time intervention, a salient feature of CBM, allows for human-model interactions. However, these interactions are prone to information leakage and can often be ineffective inappropriate communication with humans. We propose a novel uncertainty based strategy, *SIUL: Single Interventional Uncertainty Learning* to select the interventions. Additionally, we empirically test the robustness of CBM and the effect of SIUL interventions under adversarial attack and distributional shift. Using SIUL, we observe that the interventions suggested lead to meaningful corrections along with mitigation of concept leakage. Extensive experiments on three vision datasets along with a histopathology dataset validate the effectiveness of our interventional learning.

## 1 Introduction

Learning from complex relationships between real-world variables has been the key to progression of human kind [10]. While some of these relationships are now intuitive to us, some are more complex mathematical relationships. With advancement of deep learning in the field of computer vision [12], natural language processing [36], and speech recognition [4], its adoption in safety critical application is still at a nascent stage. Since neural networks are considered as black boxes and errors in applications such as medicine[35] or autonomous driving[32] may lead to catastrophe to humans. To understand the inner workings of black box neural networks, the field of Explainable AI (XAI)[3] has emerged in recent times. XAI aims to reason behind the decision making of the model. For example, if a model is presented with a set of histopathological whole slide images, we would like the model's diagnosis to be made on the basis of the tumor cell type and expect the neural network to reason itself in a way we humans can understand.

In order to explain model predictions, many efforts have recently been made. Some of these methods provide an insight into the relationships learned by the model. Post hoc based methods such as TCAV [15] carry out feature attribution of a model's result decision via directional derivatives. Saliency methods, such as grad-CAM [31], utilize first order derivatives to assign importance weights to pixels in order to visualise local explanations. Local surrogate models like LIME [29] bring explainability to black box classifiers by learning locally interpretable models around network predictions.

---

[*]Corresponding author: ivaxi-miteshkumar-sheth.1@ens.etsmtl.ca

2022 Trustworthy and Socially Responsible Machine Learning (TSRML 2022) co-located with NeurIPS 2022.

Concept based models have gained popularity due to their ability to include predefined human understandable concepts in the neural network [16]. In the context of images of animals, the concepts could be "mane" in case of a lion or "black and white stripes" in case of a zebra. Concept Bottleneck Models (CBM) essentially map input images to such interpretable concepts which in turn predicts the label. The intermediary concept prediction allows for the user to interact with the network. This interaction is facilitated by test time interventions that allows an expert to "correct" wrongly predicted concepts, possibly improving downstream predictions. Although, [24] have shown that concept representations of CBMs may result in information leakage that deteriorates the predictive performance. It is also noted that CBMs may not lead to semantically explainable concepts [25]. While there have been such attempts to discover the robustness of CBM models, we make the first attempt to study robustness of CBM under adversarial attack, in addition to distributional shifts.

Since test time interventions assist in model interaction with an expert, it is essential to create a symbiotic relationship between the model and the expert, i.e. expert learns about the correlation between a concept and its corresponding label and the model can learn true concept values from the expert. Despite being an effective technique, the current intervention procedure is highly inadequate. Therefore we propose a family of uncertainty based metrics, *SIUL: Single Interventional Uncertainty Learning* to determine the most appropriate intervention for symbiosis between model and expert.

## 2   Proposed Method

Human in loop learning has long concerned itself challenges of efficient human interaction with deep learning models. Interventions not only facilitate this interaction, it also provides explainable justifications. Often, for a medical professional, identification of key concepts is easier than identification from a plethora of diseases. While current CBMs motivate that test time intervention is beneficial, especially when an expert can correct the model, we show that this intervenability property may lead to less informative interventions. We therefore argue that there exists a symbiotic relationship between the model and user, suggesting that the oracle must query the humans "smartly" to learn the concepts to intervene on. We propose *SIUL*, a family of uncertainty based methods that proposes the concepts to be intervened on. Loosely, the method is inspired from active learning paradigm[8], although here, the concepts are not chosen iteratively.

Epistemic uncertainty is defined as the uncertainty arising due to model parameters and lack of training samples. In a realistic scenario, the availability of labelled medical data is often scarce, encouraging the application of epistemic uncertainty quantification of the concepts predicted. We use Monte-Carlo Dropout [7] in order to model epistemic uncertainty, with random dropout rate of $0.2$. We apply the dropout before prediction of concepts. For an image, $x_i$ predicting concepts $c_i...c_N$ where $N$ is the number of concepts, we evaluate $T$ softmax probabilities, $\{p_t\}_{t=1}^{T}$ for each concept prediction. We measure the uncertainty for each concept which we refer to as $\mathcal{H}(\cdot)$. The four concept uncertainty metrics we use are confidence, entropy, variance and Bhattacharyya coefficient which are defined per concept.

1. **Confidence Uncertainty** As a baseline for confidence based uncertainty quantification, we use maximum softmax probability. For each concept this can be defined as

$$U(\cdot) = conf(p) = 1 - max(p) \tag{1}$$

2. **Entropy Uncertainty** We compute the entropy based uncertainty as the measure of the expectation of the information inherited in the possible outcomes of a random variable.

$$U(\cdot) = H(p_T) = -\frac{1}{\mathbb{C}} \sum_{\mathbb{C}_i}^{\mathbb{C}} \left( \frac{1}{T} \sum_{t=1}^{T} p_{t|\mathbb{C}} \ log \frac{1}{T} \sum_{t=1}^{T} p_{t|\mathbb{C}} \right) \tag{2}$$

3. **Variance Uncertainty** Additionally, we compute the variance of all of the softmax predictions which are then averaged to give variance. uncertainty.

$$U(\cdot) = \sigma^2 = \frac{1}{\mathbb{C}} \sum_{\mathbb{C}_i}^{\mathbb{C}} \left[ \frac{1}{T} \sum_{t=1}^{T} \left( p_{t|\mathbb{C}} - \frac{1}{T} \sum_{t=1}^{T} p_{t|\mathbb{C}} \right)^2 \right] \tag{3}$$

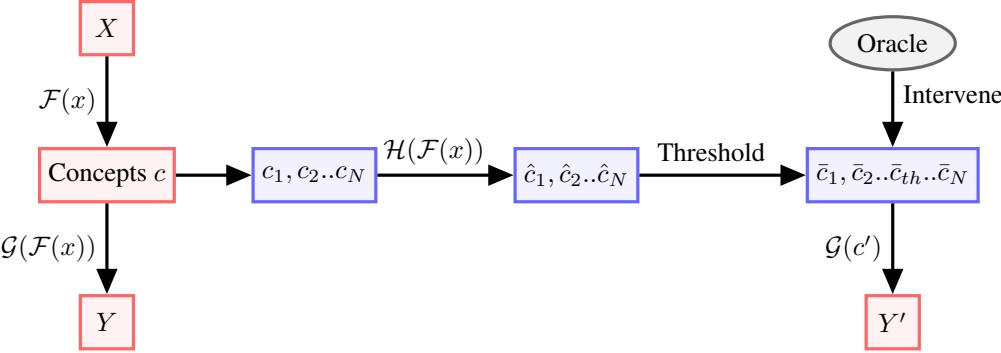

Figure 1: Our proposed concept intervention pipeline where the concepts predict uncertainty estimations that can in turn be assist the model to decide which concepts are the most appropriate to be queried

4. **Bhattacharyya coefficient** The above metrics are based on the softmax differences. It is evident that these uncertainty methods are inversely proportional to the predicted softmax probabilities. This is particularly advantageous as it is easy to apply across architechures, but since different network weights and initialisation can lead to different network outputs, we aim to use a metric that measures the overlap between the monte carlo distribution of each class. Bhattacharyya coefficient (BC) metric is particularly descriptive when there exists class overlap and low variance between the samples. We compute the BC between two classes of each concept layer. The two histograms are defined as $h^1$ and $h^2$

$$h^1 = histogram(\{p_{c_1}\}_{t=1}^T \quad \text{and} \quad h^2 = histogram(\{p_{c_2}\}_{t=2}^T \tag{4}$$

The normalised BC is then given by the predictive means of the histograms, $h^1$ and $h^2$.

$$U(\cdot) = BC(\{P\}_{t=1}^T) = \frac{1}{B} \sum_{b=1}^{B} \sqrt{h_b^1 h_b^2} \tag{5}$$

where $B$ is total number of bins.

With different uncertainty methods calculated for each attributes, we define our uncertainty over all concepts as

$$\mathcal{H}(\cdot) = [U(\cdot)_1 ... U(\cdot)_N] \quad \text{where } N \text{ is the total number of concepts} \tag{6}$$

An intervention allows the model to query the most significant concepts, in our case, it is hypothesized that the most uncertain concepts will build a symbiotic relationship between the human and the model. We therefore rank the uncertainty prediction of each concept, in order to select the most uncertain concepts. The user *thresholds* the number of concepts, it would intervene on. In contrast to current works that perform group interventions by intervening on a group of similar concepts, we perform single interventions. Group interventions require clustering of concepts on the basis of their similarity, which is not realistic as such information is not always available. Therefore single intervention are performed to minimize the dependence on priors.

## 3   Experiments

The aim of this work is to introduce strategies that a human can use to intervene on concepts. To quantify effectiveness of interventions, we compare the predictive accuracy pre and post intervention. All of our experiments use the Inception V3 [33] backbone. The results are first presented on the test sets, then on the induced distributional shift in data and a model under adversarial attack. The quantitative results presented are averaged over 10 seeds.

### 3.1   Datasets

We use natural images and medical images to test the effectiveness of interventions using our methodology. We use Caltech-UCSD Birds-200-2011 (CUB) [37] dataset for the task of bird

identification. It contains 11788 images taken from 200 different species of bird. Every image of the dataset contains 312 binary and continuous concept labels. For natural images, we additionally use Animals with Attributes 2 (AwA2)[38] dataset for the task of animal classification. The dataset contains 37322 images of 50 different species of animals. The 85 concepts in this dataset are binary. Furthermore, we create a modified MNIST [6] dataset containing handwritten digits from 0 to 9. However, the task is adapted to perform binary classification of numbers from $0 - 4$ as class 0 and $5 - 9$ as class 1. The dataset induces 10 concepts, signifying each digit.

For medical images, whole slide histopathology images were used. The tumor-infiltrating lymphocytes (TIL) [30] contains mappings based on each type of cancer. We use all 13 subsets of TCGA dataset, therefore constituting 13 cancer types. Although the popular task for such a dataset is necrosis classification, we modify the task to be a classification for different (here, thirteen) tumor types. The advantage of medical images is that their meta-data is readily available from diagnosis. A doctor uses whole slide images along with the meta-data indicating importance of fusion of different modalities. The meta data included information such as origin of tumor, age, gender, size of tumor cells which are converted to concepts.

## 3.2 Results and Analysis

In order to account for potentially incorrect interventions due to human error, we use the $95^{th}$ and $5^{th}$ percentile of the concept logit if the corresponding concept is 1 and 0 respectively. In Figure 2, we observe that random interventions, the mode of test time intervention used by current concept based models, are ineffective in facilitating optimal dialogue between human and the network. It is interesting to note that all of the proposed metrics have allowed for the model to obtain the corrected concepts in under 100 concepts on the CUB dataset. This particularly prevents correcting the redundant concepts that already have high confidence values. In the case of AwA2 dataset, despite the high accuracy, *SIUL* is able to select the appropriate concepts to converse with the human. In comparison to high label predictive performance with test time interventions in CUB and AwA2 dataset, the results on TIL dataset acknowledge the relatively less effectiveness of test time intervention. We suspect this is due to long tail of concepts with very few classes. This is particularly likely in medical applications where the diagnosis of every patient differs due to distinct ailment and its varied causes. The difference may occur due to different medical practices. Regardless, the model can learn from the *SIUL* based interventions quicker than the interventions posed by random interventions, signifying that if a doctor correct a concept, *SIUL* would be advantageous over the current strategies. It must also be noted that the accuracy by *SIUL* is marginally higher than the random interventions, credited by the generalizability property of the MC Dropout.

**Mitigating information leakage**   In addition to high accuracies, it is observed that by intervening on appropriate intermediary concepts, a known issue of leakage is mitigated. We hypothesise that concept information leakage arises due to soft concept labels. With interventions, it is apparent that soft labels are corrected by hard labels suggesting a decrease in information leakage of concept layer. However simply replacing soft labels with 0 or 1 is rather still ineffective. A layer with just hard labels loses the uncertainty information which is detrimental for generalization as we lose the predictive concept uncertainties. In our uncertainty methods, by estimating epistemic uncertainty through the Monte Carlo Dropout, this leads to soft concept prediction without containing leaked information. One may argue that addition of just MC Dropout is enough for prevention of concept leakage, but it should be noted that these are still soft prediction, therefore with interventions, hard concepts can be added without the loss of uncertainty as it has been already accounted for.

# 4   Acknowledgements

We would like to thank the Digital Research Alliance of Canada for compute resources, Google and CIFAR for research funding. IS acknowledges funding from Mitacs and Imagia Canexia Health. AAR acknowledges funding from FRQS.

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

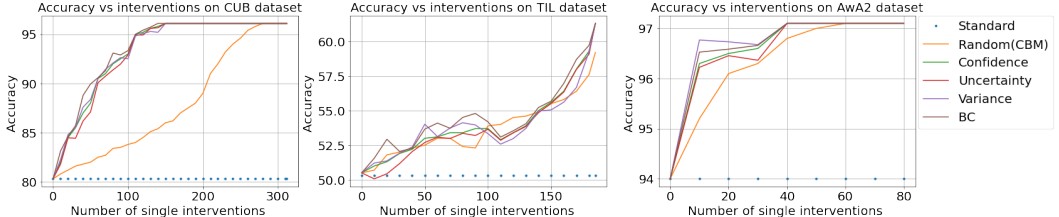

Figure 2: Accuracy vs number of intervention graphs. Due to high correlation between the concept and labels in CUB and AwA2 datasets, very high accuracy can be reached easily in comparison to TIL dataset.

# Appendix

## 5  Preliminaries

Consider a neural network that classifies an input $X \rightarrow Y$, where $X \in \mathbb{R}^n$ and $Y \in R^0$. In the generic supervised learning paradigm, we aim to learn a function $f$ that maps the the input images $X$ to label $Y$. Now, assume that concepts are present in a feature space of $C$ belonging to $\mathbb{R}^c$, where $c$ is the total number of concepts required to represent the basis of all possible values for one concept.

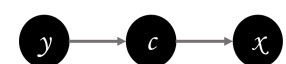

Figure 3: Causal graph for concept prediction from input images which predicts labels in turn

We also assume that there exists a hypothesis $\mathcal{G}^*_{X \rightarrow C}$ that maps the input image $x$ to concept $c$. Let $L_y$ define the loss between predicted and true targets, and $L_c$ represent the loss between predicted and true concepts. Additionally there exists a function, $\mathcal{F}^*_{C \rightarrow Y}$ that maps the concepts to the label. A model that follows the $f(x) = y$ generic supervised learning paradigm directly learns the labels from raw input images. This can be modified to the concept based learning paradigm as

$$\mathcal{F}_{C \rightarrow Y}(\mathcal{G}_{X \rightarrow C}(x)) \tag{7}$$

[16] proposes to learn the functions, $\mathcal{G}^*_{X \rightarrow C}$ and $\mathcal{F}^*_{C \rightarrow Y}$ in three ways: independent bottleneck, sequential bottleneck and joint bottleneck. For the rest of this paper, we follow the joint bottleneck paradigm since it allows for end-to-end training and has higher predictive performance.

The bottleneck model allows for interventions by editing the concept prediction. The idea of intervention is borrowed from causality, in the current regime, it is hypothesized that the images $x$ cause concepts $c$ which leads to prediction of labels $y$. The model ideally follows the causal graph presented in Figure 3. During intervention, the predicted concept $c$ can be corrected by an expert, leading to "adjusted" prediction. It is important to note that in the popular causality paradigm, interventions occur during training phase, but we consider test time interventions only i.e. the corrected concepts are not back propagated.

## 6  Related Work

Earlier work [5, 14] proposed to analyze trained models in order to induce interpretability. These post-hoc interpretations do not engage in the training paradigm to interpret model behavior. Post-hoc interpretability approaches can be divided into three main categories; sample-based [40, 13], feature-based [22, 28], and counterfactual-based methods [34, 27].

The idea of using concepts was introduced to improve model explainability. Concept-based models can be divided based on their approaches to obtaining concepts as well as their approaches to using concepts to train the models. Earlier work investigates applying human-understandable concepts in a two-stage learning process [18, 17]. While some recent studies focus on interpreting if and how a neural network uses the concept [21, 26], others study the behavior of models via generating explanations on the prediction of NN models [1, 9]. Recently [39, 19] have attempted to mitigate the

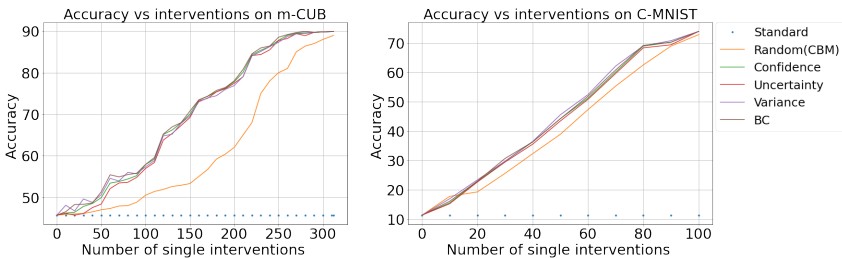

Figure 4: Accuracy vs number of interventions on synthetic CUB and MNIST datasets for out-domain setting

necessity of having human-specified concepts by automatically uncovering the concepts that models use to predict based on them.

Concept bottleneck models (CBMs) [16] proposed the benefits of intervention in models' prediction and hence increasing interpretability. CBMs requirement to access the concept labels during the training, imposes a great limitation on them. [41] proposed a new Post-hoc Concept Bottleneck (PCBM) to alleviate the limitations of CBMs by transferring the concept in the absence of concept annotation. Through the usage of fitting residuals [11] to edit the model, PCBMs improve the generalizability while reducing the spurious correlations.

# 7 More results

## 7.1 Robustness to distributional shifts

Neural networks have been show to take shortcuts and learn from spurious correlations. We posit, the reliance of shortcuts in label prediction standard models may trickle down to concept prediction in CBM. Therefore, we devise two synthetic datasets using CUB and MNIST. In MNIST dataset, we spuriously correlate the color of digit with label by the $80\%$ for the trainset, label $0$ is correlated with green color and label $1$ is correlated with red colour, loosely following [2]. For in domain testset, the spurious correlations to label are persisted whereas for the out of domain testset, the color correlations are flipped, i.e. now the label $0$ is

| Model type | CUB | | MNIST | |
|---|---|---|---|---|
| | In | Out | In | Out |
| Standard | 81.1 | 30.9 | 80.1 | 10.4 |
| Joint | 81.8 | 39.8 | 80.0 | 11.2 |

Table 1: Accuracy of standard supervised learning model and joint models under distributional shift

correlated to red color and label $1$ is correlated to green color with the probability of $90\%$. In addition to account for ambiguity and mislabelling, we additionally flip the label by $25\%$. For CUB dataset, we segment the bird images and add a colored background to all of the images. Each class, here is correlated to a randomly generated color background with probability of $80\%$ for trainset. In-domain testset contains images with similar color background probability as trainset while the correlation in out-domain test set is reduced to $10\%$.

The synthetic dataset leads the model to learn strong correlations between the background and label. From Table 1, it is evident that the joint CBMs are robust to shortcut learning in comparison to standard supervised learning models suggesting learning of concepts forces the network to predict from invariant features. We suspect that this occurs due to concept sharing for the same class and therefore, CBMs could be less robust in the case where the many of the concepts are unique such as in TIL. When a model is deployed, distributional shift is occurs, while after deployment the users may not have the resources to fine tune the model to the shift, they may still have the expert knowledge to intervene. In this case, intervention is a budget friendly, in terms of compute option to still obtain high downstream accuracy. Therefore we analyze *SIUL* based interventions under this paradigm. From Figure 4, we observe that out of all of the uncertainty methods, BC distance provides the most relevant concepts. It is evident that the effectiveness of our method, *SIUL* is dependent on the rich concept feature distribution.

## 7.2 Robustness to adversarial attacks

A challenge to deployment of machine learning models include its susceptibility to adversarial attack. An adversarial attack is essentially a small perturbation in the input data, often imperceptible to human leading in significant changes in the output of the model. The brittleness of deep learning models to such perturbations is still unknown. However, we hypothesis that predicting concepts may lead to a better insight into the black-box nature of the neural networks. A natural question arises, *would CBM be robust to adversarial attack? Additionally, how can interventions helps?*

| Model type | CUB | AwA2 |
|---|---|---|
| Standard | 62.9 | 72.0 |
| Joint | 62.3 | 71.8 |

Table 2: Accuracy of standard supervised learning model and joint models under PGD adversarial attack

In the experiment we perform non-targeted PGD [23] attack on the images of CUB and AwA2 datasets. PGD is a white box attack where a small noise proportional to the gradient of model parameters is added to the image. These changes are almost undetectable to human eye, which allows the expert to still be able to correct the concepts by visual inspection. In contrast to high accuracy obtained by CBMs, it is observed from Table 2 that CBMs do not offer a greater advantage over standard supervised learning models. From Figure 5, its is observed that the interventions do not lead to the highest accuracy as in standard model due to residue [20] in the final label prediction layer.

**Single vs Group intervention** Most of the current works group the interventions according to the similarity of concept type prediction. We argue that uncertainty of a group of concepts may be contradictory to the individual concepts present in that group. For example, for the concept, *Wing Pattern* in CUB dataset, the grouped concepts contain [has wing patter::solid, has wing patter::spotted, has wing pattern::stripped]. While the overall grouped concept wing pattern may be the most uncertain, it is possible for one of the concept to be in contrast very confident. This leads to a wasteful intervention on the confident concept. This is particularly true in the constrained i.e. fewer intervention scenario.

## 8 Conclusion

In this work we propose *SIUL*, an uncertainty based intervention selector for explainable concept bottleneck models. The proposed interventions lead to faster convergence to high accuracy. We additionally observe that the effectiveness of interventions is dependent on the nature of concepts and therefore, data. In addition, we perform robustness experiments to validate the efficacy of proposed intervention methods. We hope that this work throws a light and can motivate future work in trust worthy and explainable human in loop learning.

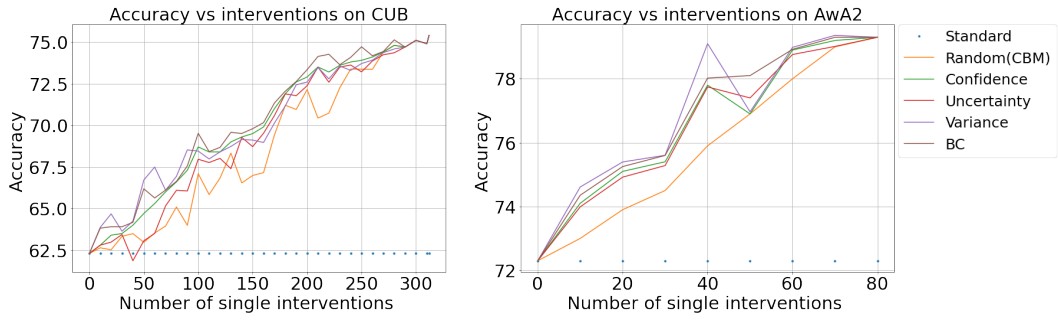

Figure 5: Interventions on CUB and AwA2 dataset post adversarial attack.

