# OpenReview forum: "Learning from uncertain concepts via test time interventions"
_NeurIPS.cc/2022/Workshop/TSRML — TSRML2022_

### Official Review · Reviewer_Qv9z · 2022-10-16
**An interesting problem and an effective solution**

**Overall Recommendation:** Learning to accept. Details are refer…
**Overall Rating:** 6

**Summary:**

Summary.

The paper is dedicated to proposing test time intervention for learning from uncertain concepts. The authors point out that the human-model interactions in CBM are prone to information leakage and can often be ineffective inappropriate communication with humans. They introduce an uncertainty-based method to select the interventions. Experiments on three vision datasets are presented.



**Strengths:**

Pros.

1. This problem setting is interesting.

2. Multiple runs of experiments are conducted.



**Weaknesses:**

Cons.

1. The arrangement is poor, which has multiple inappropriate vspace like section 4.4 and section 5.

2. Fonts in Figures 3 and 4 are too small to be recognized.

3. The experiments are limited to single network backbones. Hard to know whether it can work for other network architectures.

**Review Confidence:**

3: The reviewer is fairly confident that the evaluation is correct

---

### Official Review · Reviewer_8sBC · 2022-10-21
**This is an interesting work on enabling human-in-the-loop for "concept correction", and is well motivated from explainability point of view.**

**Overall Rating:** 7

**Summary:**

This work proposed SIUL which is an uncertainty based strategy for selecting the intervention in a Concept bottleneck model. Such CBM models work towards creating human understandable concepts, which can then be interpreted and intervened via humans to increase performance of the NNs. The proposed method is intuitive, considers 4 common uncertainty metrics for measuring uncertainty, and subsequently enabling efficient human intervention.


**Strengths:**

Enabling uncertainty-based concept selection for human intervention is of clear importance towards interpretable AI which can interact with humans. The paper is well written and easy to follow.

**Weaknesses:**

In terms of novelty, the work is limited, however, the idea is simple and intuitive.
The comparison against performance of other concept selection methods is lacking.
Last figure lacks legend.

**Overall Recommendation:**

The draft is well written and intuitive, and its publication and disclosure is beneficial to the trustworthy and explainable AI community.

**Review Confidence:**

2: The reviewer is willing to defend the evaluation, but it is quite likely that the reviewer did not understand central parts of the paper

---

### Decision · Program_Chairs · 2022-10-23

**Decision:**

Accept

**Comment:**

Interesting work on explainable AI with good empirical studies.